# LD-BFR: Vector-Quantization-Based Face Restoration Model with Latent Diffusion Enhancement

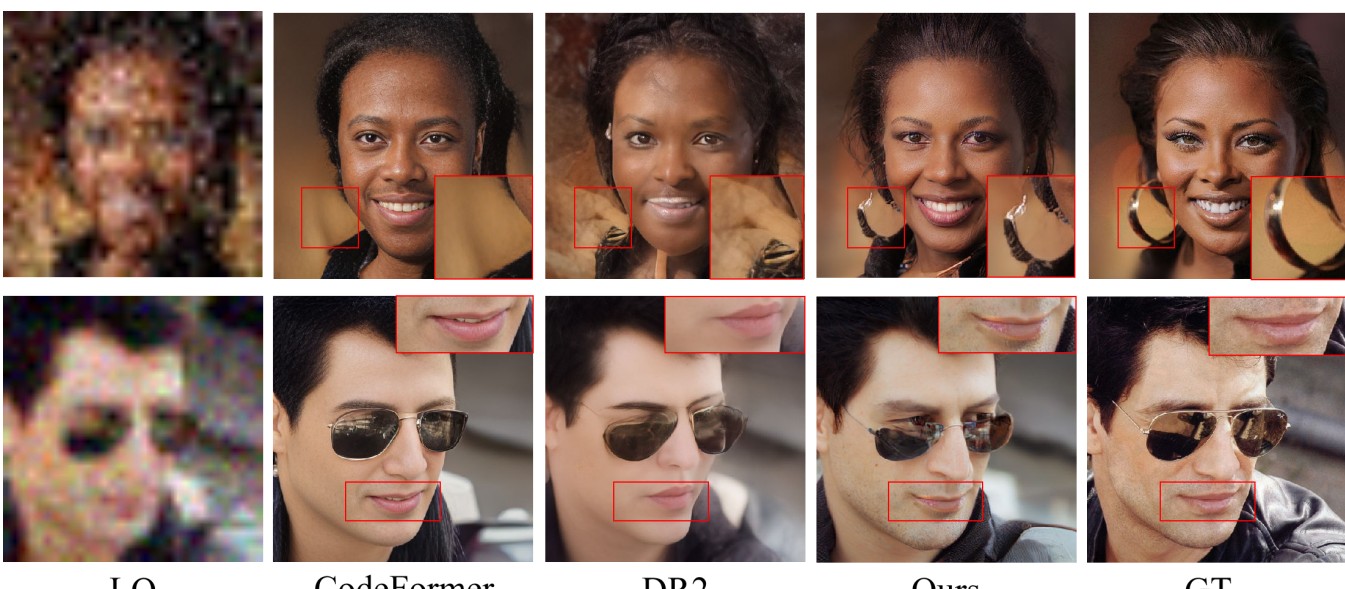

|  LQ  | CodeFormer | DR2 | Ours | GT |

**Figure 1: Comparisons of restoration quality between Codeformer, DR2 and LD-BFR. Our LD-BFR can restore high-quality facial details on various facial regions and keep the fidelity as well**

## ABSTRACT

Blind Face Restoration (BFR) aims to restore high-quality face images from low-quality images with unknown degradation. Previous GAN-based or ViT-based methods have shown promising results, but have identity details loss once degradation is severe; while recent diffusion-based methods work on image level and take a lot of time to infer. To restore images in any degradation types with high quality and spend less time, we propose LD-BFR, a novel BFR framework that integrates both the strengths of vector quantization and latent diffusion. First, we employ a Dual Cross-Attention vector quantization to restore the degraded image in a global manner. Then we utilize the restored high-quality quantized feature as the guidance in our latent diffusion model to generate high-quality restored images with rich details. With the help of the proposed high-quality feature injection module, our LD-BFR effectively injects the high-quality feature as a condition to guide the generation of our latent diffusion model. Extensive experiments demonstrate

the superior performance of our model over the state-of-the-art BFR methods.

## CCS CONCEPTS

• **Computing methodologies → Reconstruction**.

## KEYWORDS

Vector-quantization, Diffusion, Blind Face Restoration

## 1 INTRODUCTION

Blind face restoration (BFR) aims at recovering high-quality (HQ) face images from low-quality (LQ) face images with unknown degradation. Due to the ill-posed nature of this inverse problem and the diverse forms of degraded face images encountered in practice, it is highly desirable to develop a method that can faithfully restore degraded images into high-fidelity ones regardless of the type of degradation.

In recent years, there have been significant improvements in restoration quality due to the rapid development of deep learning-based methods, which can be classfied into GAN-based, ViT-based, and diffusion-based methods. GAN-based and ViT-based methods utilize various priors to guide the restoration process, including geometric [1, 11, 19], inference [3, 12, 13], and generative priors [21, 25, 29]. Besides, [5, 22, 28] use high-quality codebooks to reconstruct high-quality faces with realness and fidelity. These methods show good generation quality in most scenarios, but may

*ACM MM, 2024, Melbourne, Australia*

suffer from delicate identity features missing and hallucinating uncanny artifacts (as shown in Fig. 1). Moreover, they also suffer from the problem of training collapse and cannot deal with the dataset of long-tail distribution [15, 27]. Restored faces derived by these methods are prone to change person identities, and they are hard to achieve the balance between image restoration quality and character fidelity maintenance.

Diffusion-based methods show good results in dealing with the dataset of long-tail distribution. Some restoration methods [18, 24] leverage the pretrained diffusion model to deal with face images of known degradation forms, but are unsuitable for BFR tasks. Other diffusion-based methods [2, 15, 23] can deal with unknown degradation forms and work for BFR tasks: ILVR [2] uses pixel-wise low-frequency guidance to remove degradation; DR2 [23] further combines the diffusion model and previous BFR methods; and Diff-BFR [15] combines the cascaded diffusion model and the unconditional model. However, none of these diffusion-based BFR methods work on latent space, have a high inference efficiency, and fully utilize the information in the input low-quality images.

To achieve good performance by incorporating the advantages of both GAN-based and diffusion-based methods, we propose **LD-BFR**, a novel blind face restoration scheme based on the latent diffusion model and vector-quantized (VQ) codebook. **1)** We propose a **VQ-Restore and Diffusion-Enhancement pipeline** which integrates VQ restoration with the conditioned latent diffusion model. It employs the VQ codebook for the first-stage restoration in a global manner, and then utilizes the latent diffusion model conditioned on the high-quality quantized feature for the second-stage to enhance the quality and identity details of the output image. **2)** In the VQ-Restoration stage, we design a **Dual Cross-Attention VQ-Restoration module**. It utilizes vector quantization by a high-quality codebook and leverages the combination of Channel Cross-Attention and Spatial Cross-Attention to maintain good identity and texture information. **3)** Then, we propose a **Diffusion-based Quality Enhancement module**. Different from previous methods which utilize conditions like geometries, 3D priors to guide the restoration process, we employ the *high-quality quantized feature* obtained from the VQ-restoration module as the condition to guide a pretrained diffusion model, to further restore the image details and boost the generation quality. To inject the high-quality quantized feature into the diffusion model, we propose a novel **HQ Injection module**, which injects based on self-attention in the Encoding stage and injects based on cross-attention in the Decoding stage. It first fuses the high-quality feature with the intermediate feature in the diffusion model and then injects the fused feature into the original intermediate feature, proven to be a more effective way to inject the high-quality feature.

Extensive quantitative and qualitative experiments demonstrate that our model outperforms the existing GAN-based and diffusion-based BFR methods in different datasets. Moreover, comprehensive ablation studies validate the effectiveness of the proposed pipeline and each module. Our contributions can be summarized as the following three-fold:

- We propose LD-BFR, a novel blind face restoration framework that integrates the Vector-Quantized restoration with the conditional latent diffusion model (with the high-quality

quantized feature as a condition), which improves the restoration quality. Extensive quantitative and qualitative experiments demonstrate the superiority of our model.
- We design a Dual Cross-Attention module that consists of Spatial Cross-Attention and Channel Cross-Attention modules to get the high-quality feature. It helps fetch high-quality features as the condition for the diffusion enhancement stage.
- We propose a Diffusion-based Quality Enhancement module, which injects conditions based on self-attention in the Encoding stage and injects based on cross-attention in the Decoding stage. It further boosts the restoration ability of our LD-BFR for different degradations.

## 2 RELATED WORKS

**Blind Face Restoration (BFR).** Blind face restoration aims to address the challenge of restoring face images that suffer from complex and unknown degradation including noise, blur, low resolution, and JPEG compression artifacts. Previous works in this area can be broadly classified into two categories: prior-based methods and non-prior-based methods. Prior-based deep restoration methods can be further divided into three types: geometric [1, 11, 19, 26], inference [3, 12, 13], and generative priors [21, 25, 29].

(1) **Geometric-prior-based methods** typically leverage unique geometry and spatial distribution information of faces, *e.g.*, facial heatmaps [11], parsing maps [1, 19], and component heatmaps [26] to help restore images. However, since the geometric priors are mainly generated from degraded faces, it's difficult to obtain accurate facial priors.

(2) **Inference-based methods** [3, 13] guide the face restoration process by utilizing additional inference of the same identity as the degraded image. However, these highly requested inferences may not always be available. DFDNet [12] collects high-quality facial component features as inference priors to mitigate this problem.

(3) **Generative priors** such as pre-trained StyleGAN [10] also have been utilized to further improve restoration quality. PULUS [14] utilizes latent optimization to optimize the latent code of pre-trained StyleGAN [9]. Furthermore, GPEN [21], and GFP-GAN [25] embed generative priors into the encoder-decoder structure.

(4) **Among Non-prior based methods**, the most effective methods are ViT-based, which employ pre-trained Vector-Quantize [4, 16, 20] codebooks. Restoreformer [22], VQFR [5], and CodeFormer [28] pre-train high-quality dictionaries on entire faces.

The above methods can achieve good results, but most of them are GAN-based and ViT-based methods, they often suffer from problems of training collapse and a weak ability to deal with long-tail distribution.

**Diffusion Models.** Denoising Diffusion Probabilistic Models (DDPM) [6] show amazing results in image generation. Leveraging the rich and diverse priors offered by the diffusion model, a diffusion-based image restoration method has been proposed. SR3 [18] modifies the structure of DDPM through channel-wise concatenation to make DDPM condition on low-resolution images. ILVR [2] leverages a low-pass filter to control the generative process of pre-trained DDPM for image-translation tasks. DDRM [24] assumes the degrading process is linearly reversible and utilizes SVD (Singular Value Decomposition) to help restore images with certain degradation.

**Figure 2: The inference framework of our LD-BFR, which consists of a Dual Cross-Attention module and a Diffusion Enhancement module. In the inference stage, we first use the Dual Cross-Attention VQ-Restoration module to fetch the high-quality feature, then the Diffusion-based Quality Enhancement module restored high-quality (HQ) quantized features from the DC-VQR module to generate HQ restoration results with rich details. And finally using a Decoder to get restored images.**

However, it cannot work once the degrading matrix is unknown. DR2 [23] proposes a two-stage image restoration method. After coarsely removing degradation by the ILVR process, it leverages an enhancement module to improve the restored results. However, previous diffusion-based methods only work for known degradation removal or conduct diffusion processes on image space, which makes the inference process slow. Our method is based on the latent diffusion model and injects the HQ-feature estimated by the Encoder of VQ-GAN to guide the restoration. Since the diffusion model is good at detail enhancement and coping with long-tail distribution, we combine the VQ-GAN model with the diffusion module, using the VQ-GAN model for HQ-feature extraction, and the diffusion module for further feature enhancement. Therefore, our LD-BFR can take into account both the efficiency of reasoning and the restoration of identity details.

## 3 METHOD

### 3.1 Overview

The existing blind face restoration methods mainly employ the GAN or Diffusion model as the main framework to restore the degraded images. However, GAN-based or ViT-based models suffer from the problem of identity details lost, and diffusion models have a low inference efficiency. Few methods try to combine these two types of methods or fully utilize the information in the input low-quality images. To address these issues, we propose LD-BFR, a novel BFR framework that combines Vector-Quantized Restoration and Diffusion models, using the restored high-quality quantized feature as the bridge. The proposed model leverages the strength of vector quantization to ensure perceptual compression without massive information loss and offers high-quality texture information. It also leverages the strength of the diffusion model to help deal with long-tail distribution and further restore the identity information.

Specifically, our LD-BFR consists of two stages: **Dual Cross-Attention VQ Restoration stage (DC-VQR)**, and **Diffusion-Based Quality Enhancement stage (DQE)**. Stage 1: DC-VQR aims to extract HQ-feature by a VQ-GAN model trained on LQ-HQ pairs. The VQ-GAN takes a low-quality degraded image $x_{LQ}$ as input, initially utilizing the encoder $E_{LQ}$ to obtain the compressed

low-quality feature $f_{LQ}$. Subsequently, we acquire an intermediate high-quality feature by fetching the closest feature from the high-quality codebook of the pre-trained VQ-GAN. Then, both the intermediate high-quality feature and the low-quality feature are input into our **Dual Cross-Attention** module, which combines Channel Cross-Attention and Spatial Cross-Attention, and outputs the enhanced high-quality feature $f_{HQ}$, which encapsulates comprehensive information from the input degraded image and enables an initial restoration.

**Stage 2: DQE** aims to further boost the restoration efficacy and enhance the similarity between the identities of the input and restored images, by employing a latent diffusion model with the enhanced high-quality feature $f_{HQ}$ as a condition. In the denoising process, we first sample a random noise $z_T$ from $\mathcal{N}(0, I)$. Then, leveraging $f_{HQ}$ obtained from Stage 1 as a condition, we integrate it into the UNet model using our proposed High-Quality Feature Injection module (HQI), which injects HQ-feature based on self-attention in the Encoding stage and injects based on cross-attention in the Decoding stage. Finally, we use the Decoder from the original VQGAN of the pretrained LDM to upsample the latent feature for a high-definition and high-quality output.

In summary, the dual cross-attention Encoder first compresses the degraded face image. Then the diffusion model restores and enhances the quality and the identity details of the HQ feature. Finally, the vector-quantized Decoder upsamples the feature to get high-quality images.

### 3.2 Dual Cross-Attention VQ Restoration Module

Vector Quantization (VQ) [4] has shown its effectiveness in image inpainting, image translation, and image restoration [5, 22, 28]. The quantized codebook usually can ensure a good generation quality, since the features in the codebook are learned from high-quality real images. Therefore, we employ the vector-quantized (VQ) codebook as the basic module to restore the degraded image initially. Specifically, our Vector-Quantized Restoration module (VQR) is composed of: 1) a Low-Quality Encoder $E_{LQ}$ that encodes the low-quality degraded image $x_{LQ}$, and quantized feature fetched

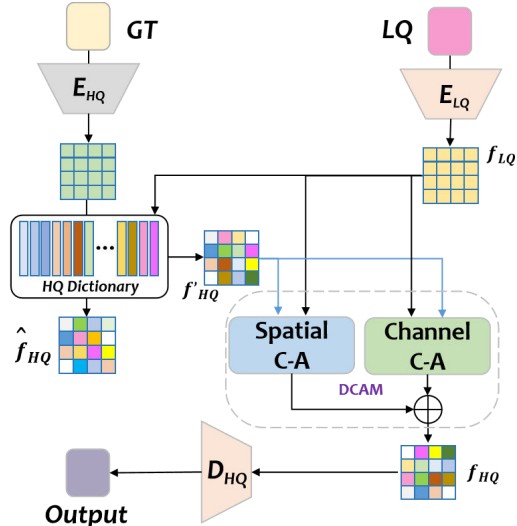

**Figure 3: The framework of Dual Cross-Attention VQ-Restoration (DC-VQR) module. It first uses VQ-Encoder and HQ-codebook to fetch feature, then leverages a combination of Channel cross-attention and Spatial cross-attention to get the final high-quality feature. The Decoder is used only for training.**

from a HQ codebook $L_{HQ}$, and 2) a quality enhancing Dual Cross-Attention Module $DCAM$ that boosts identity preservation and avoids color shift issues mentioned in previous works [28]. The Low-Quality Encoder $E_{LQ}$ and Dual Cross-Attention $DCAM$ are trained together as shown in Fig. 3.

HQ encoder, HQ decoder, HQ dictionary It should be pointed out that the codebook $L_{HQ}$ is trained on high-quality face images form FFHQ, and the Decoder used here is different from the Decoder used in the final stage of the Diffusion-Enhancement module.

**Low-Quality Encoder and Quantized Feature Fetching.** To process the input degraded image $x_{LQ}$, we introduce a perceptual encoder $E_{LQ}$ to encode the degraded image $x_{LQ}$ into low-quality feature $f_{LQ} \in R^{c \times h \times w}$, the training details is shown in Sec. 4.2. After the low-quality feature $f_{LQ}$ is obtained, we fetch the closest feature $f'_{HQ}$ from the high-quality codebook $L_{HQ}$ along dimension $h$ and $w$, i.e., fetching $h \times w$ vectors of dimension $c$ from the codebook. Since all the vectors from the codebook are learned from high-quality images, this quantized feature $f'_{HQ}$ captures the features of high-quality images and can help improve the quality of the restored images.

**Dual Cross-Attention Module.** The fetched quantized feature $f'_{HQ}$ would inevitably lose some identity information due to a large compression ratio and also have a color shift problem due to the quantization and fetching operation. To solve these issues, we introduce a Dual Cross-Attention module, which consists of the combination of Channel Cross-Attention and Spatial Cross-Attention. It takes the low-quality feature $f_{LQ}$ (for identity information and color distribution) and the fetched high-quality quantized feature $f'_{HQ}$ (for high-quality texture information) as inputs, and outputs

the refined high-quality feature $f_{HQ}$. Our dual cross-attention module is composed of a parallel spatial attention module and a channel attention module.

Specifically, in the spatial cross attention module, $f_{LQ}$ is mapped into query $Q = W_Q f_{LQ}$ and $f'_{HQ}$ is mapped into key $K = W_K f'_{HQ}$ and value $V = W_V f'_{HQ}$, and the refined feature is generated by $f_{SCA} = Softmax(\frac{Q \times K^T}{\sqrt{d}}) \times V$. In the channel cross-attention module, we get channel matrix $M_C$ from $f_{LQ}$, and the refined feature is generated by $f_{CCA} = M_C \times f'_{HQ}$. Finally, we add the two features together to get the refined high-quality feature $f_{HQ} = f_{SCA} + f_{CCA}$.

## 3.3 Diffusion-Based Quality Enhancing

With the proposed Vector-Quantized Restoration (VQR) module, our model can compress the degraded images into high-quality features, though we don't use a large compression ratio, some identity details are missing due to degradation. To restore identity details like freckles or eyelashes and solve long-tail distribution datasets, we propose to combine VQ restoration with diffusion by innovatively utilizing *restored high-quality feature* as the condition for diffusion.

Different from previous methods that utilize 3D information prior [7, 29] or identity information feature [3, 13] as the condition, our Diffusion-based Quality Enhancing module (DQE) is the first model that leverages the information restored high-quality quantized vector information (obtained in DC-VQR) as the condition, information which has the advantage of containing the features of high-quality facial details. Moreover, we propose a novel High-Quality Feature Injection module (HQI), which can better inject the high-quality feature into the latent diffusion model to restore the low-quality image compared to previous condition injection methods [17]. The main framework of our DQE is shown in Fig. 2 and the details are introduced as follows:

**HQ code-conditioned Latent Diffusion Model.** Our DQE is built on a latent diffusion model with the condition as the restored high-quality quantized feature obtained from the Vector-Quantized Restoration module $f_{HQ} = VQR(x_{LQ})$. Different from previous methods that utilize geometry or identity information as the condition, our restored HQ quantized feature $f_{HQ}$ captures the information of high-quality facial details, which can better guide the diffusion model to generate high-quality restored images.

Specifically, during training, we first encode ground truth high-quality images $\hat{x}$ into high-quality feature $\hat{f}_{HQ}$ by $E_{HQ}$(from HQ VQGAN) and add $t$-step noise to $\hat{f}_{HQ}$ in the diffusion process. Then, in the denoise process, we utilize the restored HQ quantized feature $f_{HQ} = DC-VQR(x_{LQ})$(from Compression Encoder) as the condition to guide the diffusion UNet $\mu_\theta$. We inject the restored HQ feature $f_{HQ}$ into the diffusion UNet $\mu_\theta$ with the HQ Injection module, keeping the output identity the same as the restored high-quality feature $f_{HQ}$ and boosting its quality. In the inference stage, we randomly sample a noise latent $Z_T$ from $\mathcal{N}(0, I)$ and denoise it with the condition of $f_{HQ}$ by :

$$p_\theta(z_{t-1} \mid z_t, f_{HQ}) := \mathcal{N}(z_{t-1}; \mu_\theta(z_t, f_{HQ}, t), \Sigma), \quad (1)$$

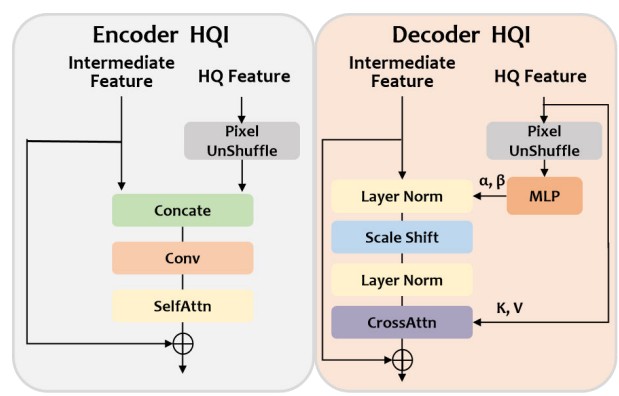

**Figure 4: The framework of our High-Quality Feature Injection module (HQI), composed of a self-attention-based HQI in the Encoding stage and a cross-attention-based HQI in the Decoding stage, which can better inject the HQ feature into the diffusion UNet model.**

where $\mu_\theta(z_t, f_{HQ}, t)$ is our HQcode-conditioned UNet model that takes both the noisy latent $z_t$ and HQ feature $f_{HQ}$ as inputs and outputs the mean of $z_{t-1}$.

**The HQ Injection module.** To inject the restored HQ feature $f_{HQ}$ into diffusion, we propose a High-Quality Feature Injection module (HQI), a simple yet effective module that can effectively inject the information into the diffusion model without influencing the original generation quality. Different from Latent Diffusion Model that solely relies on the cross-attention module to inject condition, our HQI module utilizes both cross-attention and self-attention modules for different merge purposes and better condition injection.

Our HQI module can be divided into Encoder-HQI (E-HQI) and Decoder-HQI (D-HQI). 1) **Self-Attention Based Encoder-HQI**: During the encoding stage, the input HQI firstly uses pixel unshuffle modules adjust to the condition $f_{HQ}$ and concatenate it with the intermediate feature $f_i$. Then it uses a convolution layer and a self-attention module to make sure that the texture information contained in the condition is properly merged. The output feature is $f_o = f_i + SA(Conv(Concat(f_i, PixUS(f_{HQ}))))$. 2) **Cross-Attention based Decoder-HQI:** Then in the decoding stage, we further use Decoder HQI to inject semantic and identity information from our condition $f_{HQ}$. We first use an Adaptive Layer Normalization (AdaLN) to adjust the input feature, where the scale and shift parameters of AdaLN are obtained from $MLP(f_{HQ})$. Then it uses a Cross-Attention module to merge the semantic and identity information of the condition $f_{HQ}$. The output feature is $f_o = f_i + CA(AdaLN(f_i, MLP(f_{HQ})), f_{HQ})$. During training, the module of the original diffusion model is frozen. Therefore, we do not compromise the high-quality prior-generation capability of the original diffusion model.

The reason for the design of SA-based E-HQI and CA-based D-HQI is that: During the Encoding stage, the model needs to compress the texture information and extract identity features, while the input is a noised feature initially, which necessitates us to obtain texture information from the HQ feature. Therefore, the Encoder-HQI first merges the HQ feature and the intermediate feature, and

then conducts self-attention on the merged feature to make sure the texture information in HQ feature is properly injected. In the Decoding stage, we need to avoid color shifts caused by incorrect texture combinations. Therefore, the Decoder-HQI first uses an AdaLN module, and then leverages a cross-attention module to inject semantic and identity features from the HQ feature.

**High-Quality Decoder from Pretrained VQ-GAN.** We employ the Decoder of the pretrained VQ-GAN from latent diffusion model (LDM) [17] as the upsample model after the diffusion stage. It should be noted that this VQ-GAN is different from the $E_{HQ}$ and $D_{HQ}$ in DC-VQR module.

### 3.4 Training Objectives

**Dual Cross-Attention Vector-Quantized Restoration module.** In the training process of our Vector-Quantized Restoration module, we first employ the pretrained VQGAN model [17] on the FFHQ dataset [9], including the high-quality encoder $E_{O,HQ}$, codebook $B_{HQ}$ and the decoder $D_{HQ}$. The decoder is used as our upsample decoder which can generate high-resolution images from the feature restored by the diffusion model.

Then, we train our low-quality encoder $E_{LQ}$(also it's high-quality codebook $L_{HQ}$) and the dual cross-attention module $DCAM$ on the paired data of degraded image $x_{LQ}$ and the corresponding ground truth high-quality image $\hat{x}_{HQ}$. This VQ-GAN model has a different compression ratio from the aforementioned high-quality encoder $E_{O,HQ}$, and it also has a corresponding high-quality encoder $E_{HQ}$ and HQ codebooks$L_{HQ}$. To ensure the high-quality quantized feature $f_{HQ}$ output from $DCAM$ is close to the ground truth high-quality feature $\hat{f}_{HQ} = E'_{HQ}(\hat{x}_{HQ})$. Specifically, we compute the MSE distance between $f_{HQ}$ and $\hat{f}_{HQ}$ by:

$$\mathcal{L}_{feature} = \|f_{HQ}, \hat{f}_{HQ}\|. \tag{2}$$

Moreover, we additionally add a constraint in the pixel space by computing the MSE distance of the decoded output $x_{HQ} = D'_{HQ}(f_{HQ})$ and the ground truth high-quality image $\hat{x}_{HQ}$ by:

$$\mathcal{L}_{pixel} = \|x_{HQ}, \hat{x}_{HQ}\|. \tag{3}$$

Finally, our low-quality encoder $E^*_{LQ}$ and the quality enhancing cross-attention module $DCAM^*$ (Compression Encoder module) are trained by:

$$E^*_{LQ}, DCAM^* = \underset{E_{LQ}, DCAM}{argmin} (L_{feature} + \lambda L_{pixel}). \tag{4}$$

**Diffusion-Based Quality Enhancing module.** To enhance training stability, our latent diffusion model is trained with two stages, where the first stage is trained as an unconditional model, and we add the HQ Injection module in the second stage. In the first stage, we encode the ground truth high-quality images into high-quality feature $\hat{f}_{HQ} = E_{HQ}(x\hat{}_{HQ})$, and randomly add $t$-step noise $\epsilon$ into $\hat{f}_{HQ}$ to get $z_t$, and denoise it with UNet. We minimize the distance between the added noise and the predicted noise to train the unconditional UNet model:

$$\mu_{\theta_1^*} = \underset{\mu_{\theta_1}}{argmin} \left\|\epsilon - \mu_{\theta_1}(z_t, t)\right\|_2^2, \tag{5}$$

where $\theta_1$ is the trainable parameters in the unconditional UNet model.

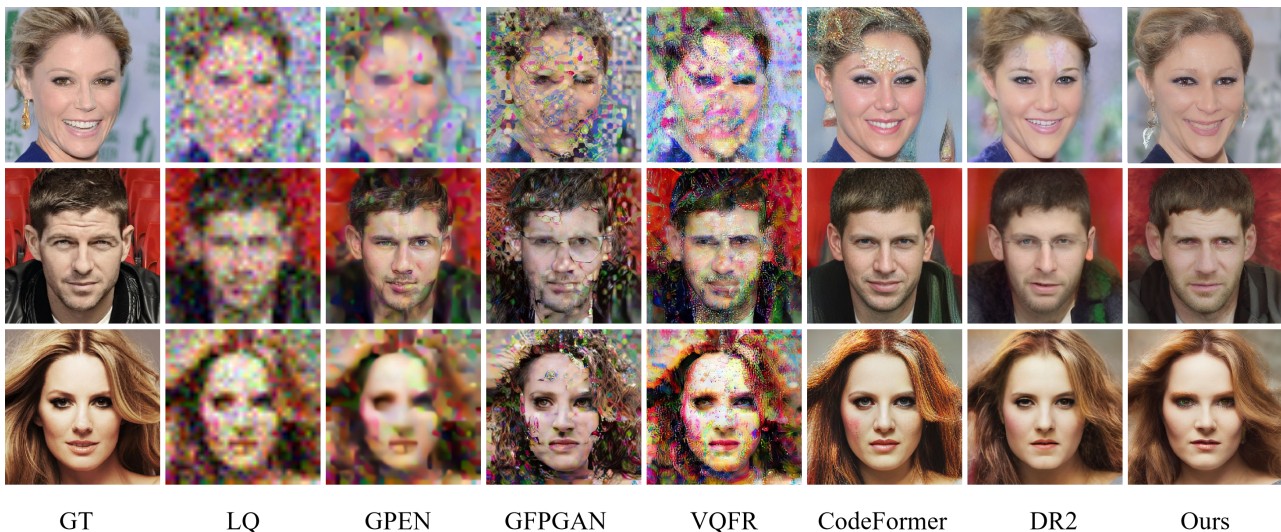

| GT | LQ | GPEN | GFPGAN | VQFR | CodeFormer | DR2 | Ours |

Figure 5: The qualitative comparison between our model and previous methods on synthetic dataset CelebA-Test.

With the unconditional UNet model being trained, our latent diffusion model can generate high-quality results by conducting the denoising process. Then, to enhance the identity similarity, we train the HQ Injection module, where we add $t$-step noise to the ground truth high-quality feature $\hat{f}_{HQ}$ to get $z_t$, and then denoise it conditioned on the restored high-quality feature $f_{HQ} = VQR(x_{LQ})$. The HQ injection module $\mu_{\theta_2}$ can be trained by:

$$\mu_{\theta^*} = \underset{\mu_{\theta_2}}{argmin} \left\| \epsilon - \mu_\theta(z_t, f_{HQ}, t) \right\|_2^2, \tag{6}$$

where $\theta_2$ is the parameters of our proposed HQ Injection module and $\theta = (\theta_1, \theta_2)$.

## 4 EXPERIMENT

### 4.1 Datasets and Implementation

**Datasets.** For the training process, we use the FFHQ dataset [9] as our training dataset which consists of 70,000 high-quality face images. We follow eq (7) to synthesize the degraded images where $\sigma, r, \delta$ and $q$ are randomly sampled from $\{0.2 : 10\}, \{1 : 8\}, \{0 : 15\}$ and $\{60 : 100\}$. And for the testing process, we evaluate our LD-BFR on one synthetic dataset (CelebA-Test), and two real-world datasets (LFW-Test and WIDER-Test). A brief introduction of each dataset is shown as below:

**(1) CelebA-Test** consists of 3,000 images and is synthesized by applying the degradation model which is commonly used in previous works [13, 21] on the testing set of CelebA-HQ images. The degrading model is as below:

$$y = \{[(x \otimes k_\sigma)\downarrow_r + n_\delta]_{JPEG_q}\} \uparrow_r . \tag{7}$$

A high-quality image $x$ is firstly blurred by a blur kernel $k_\sigma$. Then a scale factor $r$ is used to bicubically downsample the image and additive noise is added after downsampling. $n_\delta$ is the added noise and is randomly chosen from Gaussian and Poisson. Finally, applying a JPEG compression with quality factor $q$ to generate the final degraded image $y$. For a fair comparison, We follow the

Table 1: Quantitative comparisons on CelebA-Test dataset. **Bold** and underline indicates the optimal and sub-optimal performance.

| Metrics | PSNR↑ | SSIM↑ | LPIPS↓ | FID↓ |
|---|---|---|---|---|
| DFDNet | 20.1478 | 0.5333 | 0.6238 | 79.6028 |
| GPEN | 22.8403 | 0.6242 | 0.5133 | 68.0152 |
| GFPGAN | 21.3870 | 0.5287 | 0.5054 | 42.1444 |
| VQFR | 20.8036 | 0.4693 | 0.5142 | 67.6516 |
| CodeFormer | 22.4254 | 0.5934 | **0.3964** | 52.2357 |
| DR2+SPAR | 22.2827 | **0.6310** | 0.4331 | 53.7321 |
| Ours | **22.9940** | 0.6297 | 0.4282 | **38.9823** |

settings of DR2 and uniformly divide the dataset into three splits, containing mild, medium, and severe degradation settings. In this paper, for three different degradation splits, the *mild* split randomly sample $\sigma, r, \delta$ and $q$ from $\{3 : 5\}, \{4 : 4\}, \{5 : 20\}, \{60 : 80\}$, the *medium* from $\{5 : 7\}, \{8 : 8\}, \{15 : 40\}, \{40 : 60\}$ and the *severe* from $\{7 : 9\}, \{16 : 16\}, \{25 : 50\}, \{30 : 40\}$. Therefore, the synthesized LQ images dataset contains various forms of degradation. And the setting can also help us use the parameter settings of DR2 directly without further experiments.

**(2) LFW-Test.** LFW [8] consists of the first image of each person in the LFW dataset with mild degradation which contains 1711 images.

**(3) WIDER-Test.** We use the WIDER-Test dataset offered by CodeFormer [28]. It comprises 970 severely degraded face images from the WIDER Face dataset, providing a more challenging dataset to evaluate the generalizability and robustness of blind face restoration methods.

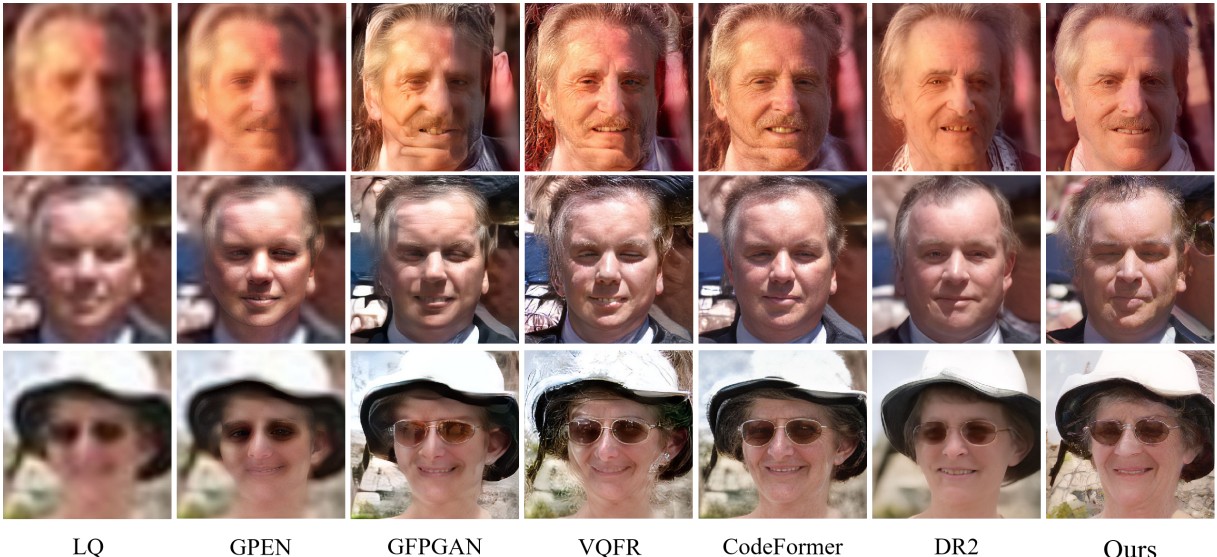

LQ      GPEN      GFPGAN      VQFR      CodeFormer      DR2      Ours

Figure 6: Comparison with state-of-the-art methods on the real-world dataset.

Table 2: Quantitative comparison with state-of-the-art methods on real face restoration with FID↓ score.

| Datasets | LFW-Test | WIDER-Test |
|---|---|---|
| DFDNet | 62.5733 | 57.8421 |
| GPEN | 54.6542 | 60.1864 |
| GFPGAN | 49.7716 | 39.5074 |
| VQFR | 50.8663 | 44.1381 |
| CodeFormer | 51.8586 | 38.7831 |
| DR2+SPAR | 45.4469 | 41.1758 |
| Ours | **38.7360** | **36.3517** |

**Metrics.** We choose pixel-level metrics PSNR, SSIM, and LPIPS as our full-reference metrics, and FID as our non-reference metric. For the quantitative experiments on the synthetic dataset, we use all four metrics. For the real-world dataset, due to the lack of reference ground-truth HQ images, we only evaluate the FID metric.

**Training details.** We employ a high-quality VQ-GAN from the pretrained LDM, and the Decoder of the VQ-GAN is used for the feature upsample after the diffusion process. The decoder is further trained for 512x512 image generation on the FFHQ dataset. For the dual cross-attention module, we chose a compression ratio of 16 and trained the corresponding high-quality VQ-GAN for HQ image generation and HQ codebooks. Then we froze the decoder and added the dual cross-attention module between the encoder and decoder, trained the Encoder and module on LQ-HQ pairs. For all stages of training, we use the Adam optimizer with a batch size of 8. We set the learning rate to $5 \times 10^{-6}$ for the training of the Encoder and Decoder. For the training of the Diffusion model, we use a smaller learning rate of $2 \times 10^{-7}$. The three stages are trained with 50K, 10K, and 200K iterations, respectively. Our method is implemented with the PyTorch framework and trained using one NVIDIA GeForce RTX 3090 GPU.

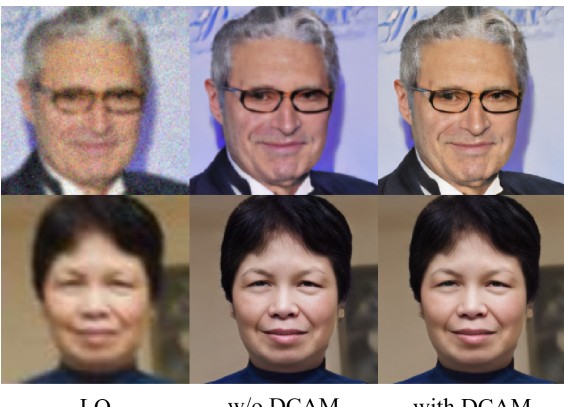

LQ      w/o DCAM      with DCAM

Figure 7: Ablation study on Dual Cross-Attention module, which can enhance the quality of fetched HQ-feature.

### 4.2 Comparisons with State-of-the-arts on BFR

We quantitatively compare our LD-BFR with six state-of-the-art face restoration methods, including DFDNet [12], GFPGAN [25], GPEN [21], VQFR [5], CodeFormer [28] and DR2 [23]. We adopt their officially released codes and models in our experiments.

**Comparison on Synthetic Dataset.** We report the quantitative comparison on the CelebA-Test on Tab. 1. Our method achieves the best scores on PSNR and FID metrics. And it also ranks second on the LPIPS and SSIM scores. This shows our outputs have closer distribution to ground truth and have high quality. Additionally, we show the qualitative comparison in Fig. 5. Once the degradation is severe, some compared methods such as GPEN [21], GFPGAN [25], and VQFR [5] cannot produce pleasant images. Although Code-Former [28] restores high-quality images, it sometimes introduces obvious artifacts or has identity errors. DR2 [23] works well on severe degradation removal, but its outputs are sometimes blurred. As

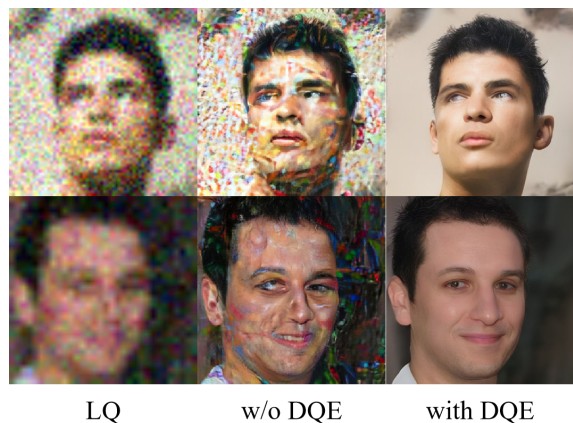

LQ          w/o DQE          with DQE

**Figure 8: Ablation study on DQE. The diffusion model can enhance the quality of the restored images.**

is shown in Fig. 5, our approach is very good at maintaining some identity details, such as earrings and jewelry which are ignored by the GAN-based method. Due to the combination of VQ-GAN and Diffusion model via the restored HQ feature, our method not only improves the details of the restored face images but also is good at dealing with severe degradation.

**Comparison on Real-world Dataset.** As is shown in Tab. 2, our method achieves the highest scores of FID on the LFW-Test and WIDER-Test datasets. This shows that LD-BFR can handle real-world datasets with different degradation. From the visual comparisons in Fig. 6, it can be found that our method is robust to real severe degradation and produces visually pleasant results.

## 4.3 Ablation Study

**Importance of dual Cross-Attention module.** To verify the effectiveness of the dual Cross-Attention module, we train a LQ Encoder without dual cross-attention and a LQ encoder with dual cross-attention under the same settings. To avoid the influence of the Diffusion model, we test both encoders without the diffusion process and use the same decoder and codebook. The visual comparison is shown in Fig. 7. The results indicate that the dual cross-attention module effectively enhances the restoration quality and identity preservation. It solves the color shifts problem of restored images.

**Effectiveness of Diffusion-based Quality Enhancing.** We investigate the effectiveness of the Diffusion process on the image with severe degradation. We show the qualitative comparison in Fig. 8, where the results indicate that diffusion enhancement is the key to ensuring the robustness and effectiveness of our method. It quality-enhancing process further removes degradation and enhances the restored feature. The restored face has severe artifacts without the diffusion process (DQE) once the degradation is severe.

**Effectiveness of our HQI settings.** To show the effectiveness of our input HQI and output HQI, we trained four models under the same conditions. These four models use the same VQ model to compress features and upsample, they all use the U-Net structure but use different methods to inject features. Their structures are shown in Fig. 9: model **A** uses concat module to inject feature,

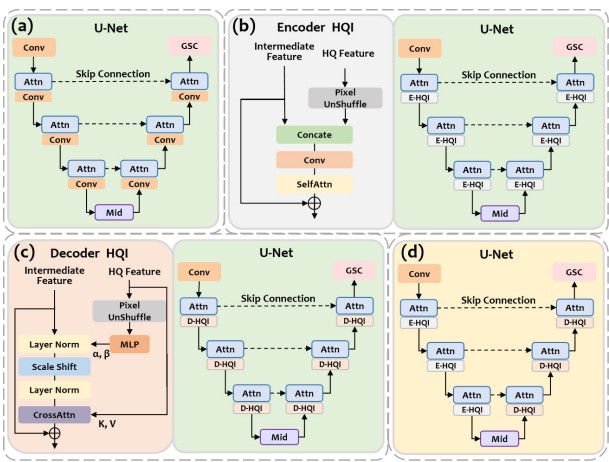

**Figure 9: The different structures compared in ablation study on HQI module.**

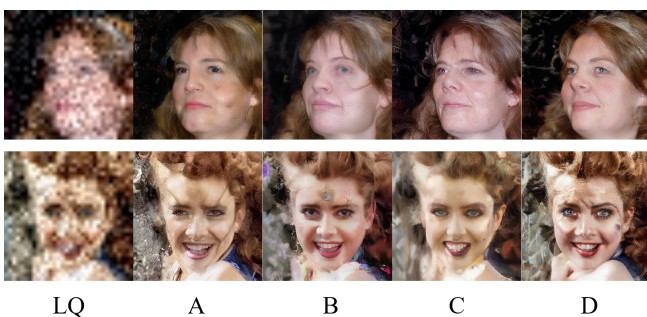

LQ          A          B          C          D

**Figure 10: Ablation qualitative study on HQI module. A D correspond to the restored images of structures in Fig. 9.**

model **B** uses Encoder HQI in both Encoding and Decoding stages, model **C** uses Decoder HQI in both stages, while model **D** is our HQI framework.

As shown in Fig. 10, model A simply uses the conv layer to inject the condition. Its restored images have some unbearable artifacts and have identity details lost. Model B only uses self-attention-based Encoder-HQI to inject the condition, its restored images suffer from the problem of color shift, which may be caused by unseasonable texture information merge. Model C only uses cross-attention-based Decoder-HQI to merge the condition, it keeps the high-quality details but changes the fidelity. From the visualization, our HQI helps accurately merge and restore high-quality facial details to better match the degraded input.

## 5 CONCLUSION

In this paper, we propose LD-BFR, a novel framework integrating both the strengths of vector quantization and latent diffusion, which can restore images with high quality. By leveraging vector quantization, we can ensure a good generation quality of the restored image. Moreover, we introduce a latent diffusion conditioned on the quantized feature by our high-quality injection module, which further improves the generation quality and boosts identity restoration. Extensive experiments demonstrate the superior performance of our LD-BFR over the existing methods.

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
