# OpenReview forum: "LD-BFR: Vector-Quantization-Based Face Restoration Model with Latent Diffusion Enhancement"
_acmmm.org/ACMMM/2024/Conference — MM2024 Poster_

### Official Review · Reviewer_Aso8 · 2024-05-21

**Rating:** 4
**Confidence:** 2

**Summary:**

This is a work on BFR that utilizes VQGAN to do preliminary processing in the latent space and further repair in the latent space using LDM. In the preliminary processing of VQGAN, the training model of VQGAN is directly used, inheriting the Encoder and Codebook parts. The Dual Cross-Attention vector quantization is proposed to combine the details of the dictionary thrower and the ID information of the LQ as the final guidance of the LDM. In the process of further restoration of the latent space by LDM, compared with the conditional embedding approach in the past, the paper proposes the high-quality feature injection module, which uses different embedding methods in the downsampling and upsampling layers of U-Net. This article is clearly structured and descriptive, and puts forward its own ideas about both guidance content and embedding of DMs, and verifies the feasibility by comparing them in the experimental results.

**Strengths:**

(1) The content description of the paper is very clear, and the detailed illustrations of each stage are also very clear.

(2) In the content setting of guidance, it tries to use the way of dictionary-based preprocessing, which is more accurate for the restoration of excessively degraded face images.

(3) In the embedding part of conditional, it innovatively puts forward the scheme with different embedding forms in the up-sampling and down-sampling stages, but it does not specifically analyze and explain the feasible reasons in it.

**Limitations:**

(1) The abstract mentions “To restore images in any degradation types with high quality and spend less time”, may I ask which aspect does the less time here specifically refer to, and the article does not follow up with a corresponding find.

(2) There is no comparison between Restoreformer and DiffBIR in experiments, which are the two more classical comparison methods.

(3) Methods such as GFPGAN, VQFR, etc. have been done on CelebChild, a real-world dataset, can you provide the experimental results on this dataset?

(4) Can you provide the quantitative comparison results of the ablation experiments on the CelebA dataset to see the extent of the role of each module from a quantitative point of view？

(5) In Figure 9 and Figure 10 when the ablation comparison on the HQI module is given, is there any quantitative comparison of the results of this part, or the convergence plot of the training loss?

(6) The channel cross-attention in DC-VQR, why is it set up like that in the article? What does $W*f$ mean?

(7) How is the whole model trained here? Because the model has two phases and the output of the first phase is conditional on the second phase.

**Suitability:**

3

---

### Official Review · Reviewer_2GK9 · 2024-05-24

**Rating:** 2
**Confidence:** 3

**Summary:**

The paper introduces a novel Blind Face Restoration (BFR) framework called LD-BFR, which aims to restore high-quality face images from low-quality images with unknown degradation. The existing GAN-based or ViT-based methods have shown promising results but suffer from identity details loss when the degradation is severe, while recent diffusion-based methods are time-consuming. LD-BFR integrates vector quantization and latent diffusion to address these issues.

**Strengths:**

(+) The idea of Integration of vector quantization and latent diffusion for efficient face restoration.

**Limitations:**

(-) The author concludes this method can deal with long-Tari distribution and inference efficiency problems by combining VQ and LDM. However, this cannot support the paper's motivation convincedly. Some methods have already been introduced for BFR [1,2,3]. Besides. There are already numerous works in FR with VQ+LDM-based methods like [4].

(-) Although the author provides some visual results of DCAM/DQE/HQI modules, they seem just brown from some existing works. Besides, no quantitative comparison results are provided to support its effectiveness. The results are not convincing. To me, this work simply replaces the previous VQFR with an LDM without any other insightful designations.

[1] Efficient Diffusion Model for Image Restoration by Residual Shifting

[2] Towards Real-World Blind Face Restoration with Generative Diffusion Prior

[3] CLR-Face: Conditional Latent Refinement for Blind Face Restoration Using Score-Based Diffusion Models

[4] Image Super-Resolution via Latent Diffusion: A Sampling-Space Mixture of Experts and Frequency-Augmented Decoder Approach

Additional suggestions:
 Line#114-123. The author should not include diffusion-based methods (#Line110).
Fig.9 and Fig.10 are confusing. It’s hard for reader to link the relations between two figures.
Fig.10 Caption A-D

**Suitability:**

2

---

### Official Review · Reviewer_R8TZ · 2024-05-24

**Rating:** 3
**Confidence:** 4

**Summary:**

The paper proposes a blind face restoration framework (LD-BFR) combining vector quantization and latent diffusion. Specifically, LD-BFR first restores degraded face images globally using the proposed dual cross-attention vector quantization, then generates high-quality images with rich details through a latent diffusion model guided by restored quantized features. The proposed high-quality feature injection module enhances restoration quality. The experiments show that the proposed method achieves competitive performance.

**Strengths:**

1. The paper is overall well-written.
2. The proposed method achieves competitive performance in face restoration tasks.

**Limitations:**

1. Unclear motivation and lack of insights. The proposed method is a combination of the vector quantization technique and the diffusion model. The vector quantization technique has been extensively studied in the CodeFormer and VQFR.
2. Novelty. In the proposed method, the core module is the proposed dual cross-attention. However, channel and spatial attention are very common in recent literature.
3. Lacking comparison methods. Recent diffusion methods like DiffBFR [1] and StableSR [2] are not used for comparison.
   [1]. Lin et al., DiffBIR: Towards Blind Image Restoration with Generative Diffusion Prior
   [2]. Wang et al., Exploiting Diffusion Prior for Real-World Image Super-Resolution.

4. As a diffusion-based method, the efficiency is important in applications. However, the authors do not provide the running efficiency comparison among different methods.

**Suitability:**

2

---

### Official Review · Reviewer_XBwL · 2024-05-24

**Rating:** 4
**Confidence:** 2

**Summary:**

The paper proposes LD-BFR for Blind Face Restoration (BFR) task that combines vector quantization and latent diffusion to restore high-quality facial images from low-quality inputs with unknown degradation. LD-BFR employs a Dual Cross-Attention vector quantization module for global restoration, followed by a latent diffusion model enhanced with high-quality quantized features for detailed image generation. This approach aims to overcome the limitations of existing GAN-based and diffusion-based methods, such as identity detail loss and inefficiency. Extensive experiments demonstrate the superior performance of LD-BFR in maintaining fidelity and quality across various degradation types.

**Strengths:**

This paper is well-organized and straightforward. The quantitative and qualitative experiment results clearly demonstrate the advantages of the proposed pipeline, particularly showing the benefits of the Dual Cross-Attention VQRestoration (DC-VQR) module. The experiments are not only conducted on the common CelebA test dataset, but also on the LFW-Test and WIDER-Test datasets to show the robustness of this method. Furthermore, the sufficient ablation study of the HQI module shows the effectiveness of the structure design.

**Limitations:**

[1]. Why does the PSNR of GPEN reach 22.8 in Table 1? The qualitative results of GPEN on the CelebA-Test look not good enough.

[2]. Are there any quantitative results of the ablation study? The comprehensive quantitative result of each module can better prove the effectiveness of your structures.

[3]. Could you show the GT on the ablation study for better comparison?

[4]. What's the step number of the denoising process during the inference? How does the step number affect your final visual results? Could you show the inference time of different methods? Is your method much slower than the non-diffusion-based model?

**Suitability:**

2

---

### Meta-Review · Area_Chair_SD9D · 2024-07-01

**Recommendation:** Accept (Poster)
**Confidence:** 5

**Metareview:**

This work combines vector quantization and latent diffusion models for face restoration, which achieves state-of-the-art performance. Initially, this paper received two borderline accept, one borderline reject, and one weak reject. After the rebuttal, all the reviewers were happy with the authors' response, and the final ratings are three borderline accept and one weak accept. There are clear technical merits of this paper, and we would recommend acceptance to this paper.